# DMin: Scalable Training Data Influence Estimation for Diffusion Models

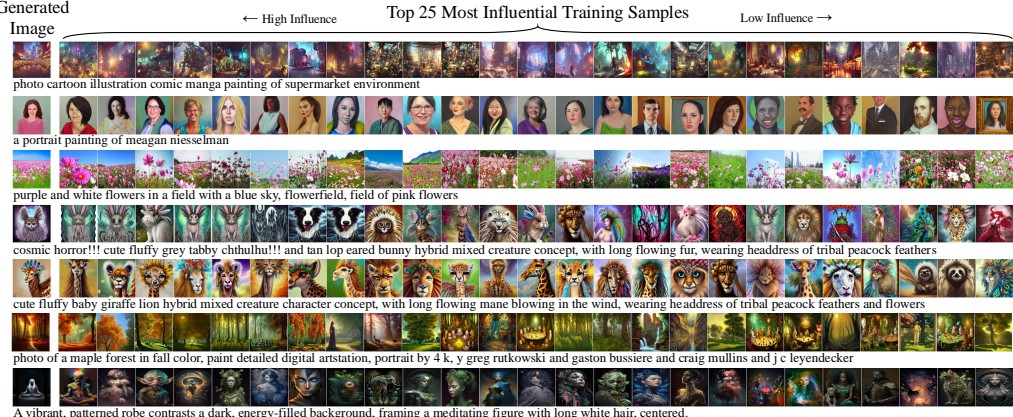

Figure 1: Examples of influential training samples, with prompts displayed below generated image. (SD 3 Medium with LoRA, $v = 2^{16}$).

## ABSTRACT

Identifying the training data samples that most influence a generated image is a critical task in understanding diffusion models (DMs), yet existing influence estimation methods are constrained to small-scale or LoRA-tuned models due to computational limitations. To address this challenge, we propose DMin (**D**iffusion **M**odel **in**fluence), a scalable framework for estimating the influence of each training data sample on a given generated image. To the best of our knowledge, it is the first method capable of influence estimation for DMs with billions of parameters. Leveraging efficient gradient compression, DMin reduces storage requirements from hundreds of TBs to mere MBs or even KBs, and retrieves the top-$k$ most influential training samples in under 1 second, all while maintaining performance. Our empirical results demonstrate DMin is both effective in identifying influential training samples and efficient in terms of computational and storage requirements.

## 1 INTRODUCTION

Diffusion models have emerged as powerful generative models, capable of producing high-quality images and media across various applications (Croitoru et al., 2023; Yang et al., 2024a; Luo, 2022; Zhang et al., 2023). Despite their impressive performance, the extremely large scale and complexity of the datasets used for training are often sourced broadly from the internet (Schuhmann et al., 2022; Wang et al., 2023; Srinivasan et al., 2021). This vast dataset diversity allows diffusion models to generate an range of content, enhancing their versatility and adaptability across multiple domains (Li et al., 2023; Chen et al., 2023). However, it also means that these models may inadvertently generate unexpected or even harmful content, reflecting biases or inaccuracies present in the training data.

This raises an important question: *given a generated image, can we estimate the influence of each training data sample on this image?* Such an estimation is crucial for various applications, such as understanding potential biases (Kong et al., 2022; Lyu et al., 2023) and improving model transparency by tracing the origins of specific generated outputs (Koh & Liang, 2017; Choe et al., 2024; Grosse et al., 2023).

Recently, many studies have explored influence estimation in diffusion models (Mlodozeniec et al., 2024; Kwon et al., 2024; Mlodozeniec et al., 2024; Ogueji et al., 2022; Georgiev et al., 2023). These methods assign an influence score to each training data sample relative to a generated image, quantifying the extent to which each sample impacts the generation process. For instance, DataInf (Kwon et al., 2024) and K-FAC (Mlodozeniec et al., 2024) are influence approximation techniques tailored for diffusion models. However, they are both second-order methods that require the inversion of the Hessian matrix. To approximate this inversion, they must load all the gradients of training data samples across several predefined timesteps. Notably, in the case of the full-precision Stable Diffusion 3 medium model (Esser et al., 2024), the gradient of entire model requires approximately 8 GB of storage. Collecting gradients for one training sample over 10 timesteps would consume $8 \times 10 = 80$ GB. Scaling this requirement to a training dataset of $10,000$ samples results in a storage demand of around 800 TB – far exceeding the capacity of typical memory or even hard drives. Given that diffusion models are often trained on datasets with millions of samples, this storage demand becomes impractical. Consequently, these methods are limited to LoRA-tuned models or small diffusion models (Ho et al., 2020; Rombach et al., 2022). Although some prior works have applied gradient compression, such as SVD (Grosse et al., 2023) and quantization (Mlodozeniec et al., 2024), the achieved compression rates are insufficient to maintain performance at this scale.

Alternatively, Journey-TRAK (Georgiev et al., 2023) and D-TRAK (Ogueji et al., 2022) are first-order methods for influence estimation on diffusion models, which are extended from TRAK (Park et al., 2023) on deep learning models. Both approaches utilize random projection to reduce the dimensionality of gradients. However, for large diffusion models, such as the full-precision Stable Diffusion 3 Medium model, the gradient dimensionality exceeds 2 billion parameters. Using the suggested projection dimension of $32,768$ in D-TRAK, store a such $2B \times 32,768$ projection matrix requires more than 238 TB of storage. Even projection matrix is dynamically generated during computation, the scale of these operations substantially slows down the overall process. As a result, they are only feasible for small models or adapter-tuned models.

**Challenges.** Although these approaches have demonstrated superior performance on certain diffusion models, there are several key challenges remain: **(1) Scalability on Model Size:** Existing methods either require computing second-order Hessian inversion or handling a massive projection matrix, both of which restrict their applicability to large diffusion models. **(2) Scalability on Dataset Size:** Diffusion models frequently rely on datasets containing millions of samples, making the computation of a Hessian inversion for the entire training dataset impractical. Additionally, storing the full gradients for all training data samples presents a significant challenge. **(3) Fragility of Influence Estimation:** Previous studies have demonstrated that the fragility of influence estimation in extremely deep models (Lin et al., 2024; Basu et al., 2021; Epifano et al., 2023; Ghorbani et al., 2019). Similarly, we observed this fragility in large diffusion models, regardless of whether they use U-Net or transformer.

To address these challenges, in this paper, we propose `DMin`, a scalable influence estimation framework for diffusion models. Unlike existing approaches that are limited to small models or LoRA-tuned models, the proposed `DMin` scales effectively to larger diffusion models with billions of parameters. For each data sample, `DMin` first computes and collects gradients at each timestep, then compresses these gradients to MBs or KBs while maintaining performance. Following this compression, `DMin` can accurately estimate the influence of each training data sample on a given generated image or retrieve the top-$k$ most influential samples on-the-fly using K-nearest neighbors (KNN) search, enabling further speedup based on the specific task.

**Contributions.** The main contributions of this paper are:

- We introduce `DMin`, a scalable influence estimation framework for DMs, compatible with architectures, from small models and LoRA-tuned models to models with billions of parameters.
- To overcome storage and computational limitations, `DMin` employs a gradient compression technique, reducing storage from around 40 GB to 80 KB per sample while maintaining accuracy, enabling feasible influence estimation on large models and datasets.
- `DMin` utilizes KNN to retrieve the top-$k$ most influential training samples for a generated image on-the-fly.
- Our experimental results confirm `DMin`'s effectiveness and efficiency in influence estimation.
- We provide an open-source PyTorch implementation with multiprocessing support[1].

---

[1] https://anonymous.4open.science/r/DMin

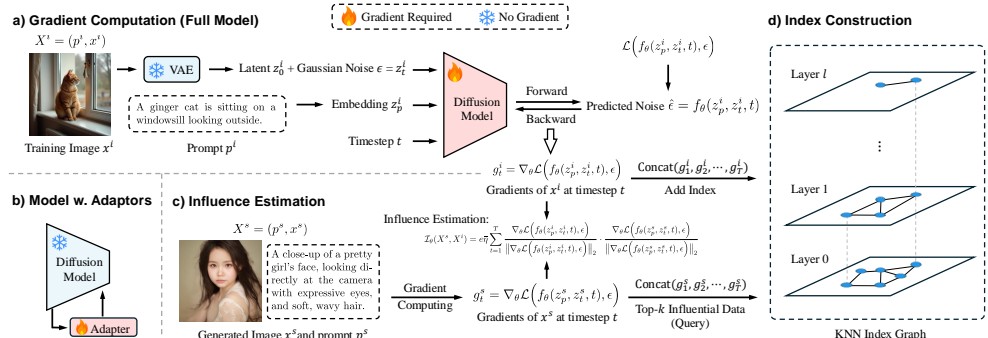

Figure 2: Overview of the proposed DMin. **(a)** In gradient computation, given a training data sample (a pair of prompt $p^i$ and image $x^i$) and a timestep $t$, the data passes through the diffusion model in the same manner as during training. After the backward pass, the gradients $g_t^i$ at timestep $t$ can be obtained. **(b)** For the full model, gradients are collected from the UNet or transformer, whereas for models with adapters, such as LoRA, gradients are collected only from the adapter. **(c)** For a prompt $p^s$ and the corresponding generated image $x^s$, the gradients are obtained in the same way as in Gradient Computation. The influence $\mathcal{I}_\theta(X^s, X^i)$ is then estimated by aggregating gradients across timesteps from $t = 1$ to $T$. **(d)** In some cases, only the influential data samples are needed; in such instances, KNN can be utilized to retrieve the top-$k$ most influential samples within seconds.

## 2 INFLUENCE ESTIMATION FOR DIFFUSION MODELS

For a latent diffusion model, data $x_0$ is first encoded into a latent representation $z_0$ using an encoder $E$ by $z_0 = E(x_0)$. The model then operates on $z_0$ through a diffusion process to introduce Gaussian noise and iteratively denoise it. The objective is to learn to reconstruct $z_0$ from a noisy latent $z_t$ at any timestep $t \in \{1, 2, \cdots, T\}$ in the diffusion process, where $T$ is the number of diffusion steps. Let $\epsilon_t \sim \mathcal{N}(0, I)$ denotes the Gaussian noise added at timestep $t$. We define the training objective at each timestep $t$ as follow:

$$\theta^* = \arg\min_\theta \mathbb{E}_{z_0, t}\left[\mathcal{L}\Big(f_\theta(z_t, t), \epsilon_t\Big)\right] \tag{1}$$

where $\theta$ represents the model parameters, $z_t$ is the noisy latent representation of $z_0$ at timestep $t$, $f_\theta(z_t, t)$ represent the model's predicted noise at timestep $t$ for the noisy latent $z_t$. $\mathcal{L}(\cdot)$ is the loss function between the predicted noise and actual Gaussian noise.

Given a test generation $x^s$, where $x^s$ is generated by a well-trained diffusion model with parameters $\theta^*$, the goal of influence estimation is to estimate the influence of each training data sample $x^i (1 \le i \le N)$ on generating $s$, where $N$ is the size of training dataset. Let $z_t^i$ represents the latent representation of $x^i$ on timestep $t$, $z_t^s$ denotes the latent representation of test generation $s$ on timestep $t$.

Considering in the $\alpha$-th training iteration, the model parameter $\theta_{\alpha+1}$ are updated from $\theta_\alpha$ by gradient descent on the noise prediction loss for batch $B = (B_z, B_t, B_\epsilon)$:

$$\theta_{\alpha+1} = \theta_\alpha - \eta_\alpha \frac{1}{|B|} \sum_{(z_t, t, \epsilon_t) \in B} \nabla_{\theta_\alpha} \mathcal{L}(f_{\theta_\alpha}(z_t, t), \epsilon_t) \tag{2}$$

where $\eta_\alpha$ denotes the learning rate in the $\alpha$-th iteration, $(z_t^i, t^i, \epsilon_t^i) \in B$, and the contribution of $(z_t^i, t^i, \epsilon_t^i)$ to batch gradient is $\frac{1}{|B|} \nabla_{\theta_\alpha} \mathcal{L}(f_{\theta_\alpha}(z_t^i, t^i), \epsilon_t^i)$. The influence of this training iteration $z_t^i$ with respect to $z_t^s$ on timestep $t$ can be quantified as the change in loss:

$$\mathcal{I}_{\theta_{\alpha+1}, t}(x^s, x^i) = \mathcal{L}\Big(f_{\theta_\alpha}(z_t^s, t), \epsilon_t^i\Big) - \mathcal{L}\Big(f_{\theta_{\alpha+1}}(z_t^s, t), \epsilon_t^i\Big) \tag{3}$$

where $z_t^s = E(x^s) + \epsilon_t^i$, denoting the latent representation of $s$ adding the Gaussian noise $\epsilon_t^i$, and $\mathcal{I}_{\theta_{\alpha+1}, t}(x^s, x^i)$ represents the influence of $x^i$ with respect to $s$ at the $\alpha$-th iteration with timestep $t$. Then $\mathcal{L}\Big(f_{\theta_{\alpha+1}}(z_t^s, t), \epsilon_t^i\Big)$ can be expanded by Taylor expansion:

$$\mathcal{L}\Big(f_{\theta_{\alpha+1}}(z_t^s, t), \epsilon_t^i\Big) = \mathcal{L}\Big(f_{\theta_\alpha}(z_t^s, t), \epsilon_t^i\Big) + (\theta_{\alpha+1} - \theta_\alpha)\nabla_{\theta_\alpha}\mathcal{L}\Big(f_{\theta_\alpha}(z_t^s, t), \epsilon_t^i\Big) + O(||\theta_{\alpha+1} - \theta_\alpha||^2)$$

Given the small magnitude of the learning rate $\eta$, we disregard the higher-order term $O(||\theta_{\alpha+1} - \theta_\alpha||^2)$, as it scales with $O(||\eta_\alpha||^2)$ and is therefore negligible. Then we have:

$$\mathcal{I}_{\theta_{\alpha+1},t}(x^s, x^i) = \mathcal{L}\Big(f_{\theta_{\alpha+1}}(z_t^s, t), \epsilon_t^i\Big) - \mathcal{L}\Big(f_{\theta_\alpha}(z_t^s, t), \epsilon_t^i\Big) \tag{4}$$

$$\Rightarrow \eta_\alpha \nabla_{\theta_\alpha} \mathcal{L}\Big(f_{\theta_\alpha}(z_t^i, t), \epsilon_t^i\Big) \nabla_{\theta_\alpha} \mathcal{L}\Big(f_{\theta_\alpha}(z_t^s, t), \epsilon_t^i\Big)$$

To estimate the influence of training data sample $x^i$, we can summing up all the training iteration training on $x^i$ and timesteps on $t \in \{1, 2, \cdots, T\}$:

$$\mathcal{I}_{\theta*}(x^s, x^i) = \sum_{\theta_a:x^i} \sum_{t=1}^{T} \eta_\alpha \nabla_{\theta_\alpha} \mathcal{L}\Big(f_{\theta_\alpha}(z_t^i, t), \epsilon_t^i\Big) \nabla_{\theta_\alpha} \mathcal{L}\Big(f_{\theta_\alpha}(z_t^s, t), \epsilon_t^i\Big)$$

where $\theta_a{:}x^i$ denotes the iteration training on $x^i$. However, it is impractical to store model parameters and the Gaussian noise for each training iteration. Thus, for a diffusion model with parameter $\theta$, given a test generation $s$, we estimates the influence of a training data sample $x^i$ with respect to test generation $s$ by:

$$\mathcal{I}_\theta(x^s, x^i) = e\bar{\eta} \sum_{t=1}^{T} \nabla_\theta \mathcal{L}\Big(f_\theta(z_t^i, t), \epsilon\Big) \nabla_\theta \mathcal{L}\Big(f_\theta(z_t^s, t), \epsilon\Big) \tag{5}$$

where $e$ is the number of epochs, $\bar{\eta}$ is the average learning rate during training, $\epsilon$ corresponds to the Gaussian noise used in the training process. However, storing all Gaussian noise from the training process is impractical. Therefore, we randomize the Gaussian noise following the same distribution as in the training process for influence estimation.

Similarly, for a text-to-image model, the influence of a training data sample $X^i = (p^i, x^i)$ with respect to test generation $X^s = (p^s, x^s)$ can be estimated by:

$$\mathcal{I}_\theta(X^s, X^i) = e\bar{\eta} \sum_{t=1}^{T} \nabla_\theta \mathcal{L}\Big(f_\theta(z_p^i, z_t^i, t), \epsilon\Big) \nabla_\theta \mathcal{L}\Big(f_\theta(z_p^s, z_t^s, t), \epsilon\Big) \tag{6}$$

where $p^i$ is the prompt of training data sample, $p^s$ denotes the prompt of test generation, $z_p^i$, $z_p^s$ is the embedding of prompt $p^i$ and $p^s$, respectively.

## 3  DMIN: SCALABLE INFLUENCE ESTIMATION

For a given generated image $x^s$ and the corresponding prompt $p^s$, the objective of DMin is to estimate an influence score $\mathcal{I}_\theta(X^s, X^i)$ for each training pair $X^i = (p^i, x^i)$, where $X^s = (p^s, x^s)$. Based on Equation 6, the $\mathcal{I}_\theta(X^s, X^i)$ can be expressed as the summation of the inner product between the loss gradients of the training sample and the generated image, computed with respect to the same noise $\epsilon$ across timesteps $t \in \{1, 2, \cdots, T\}$. Since the training dataset is fixed and remains unchanged after training, a straightforward approach is to cache or store the gradients of each training sample across timesteps. When estimating the influence for a given query generated image, we only need to compute the gradient for the generated image and perform inner product with the cached gradients of each training sample.

However, as the size of diffusion models and training datasets grows, simply caching the gradients becomes infeasible due to the immense storage requirements. For instance, for a diffusion model with 2B parameters and $1,000$ timesteps, caching the loss gradient of a single training sample would require over $7,450$ GB of storage, making the approach impractical when scaled to large datasets.

In this section, we explain how we reduce the storage requirements for caching large gradients from gigabytes to kilobytes (Gradient Computation) and how we perform influence estimation for a generated image on the fly (Influence Estimation), as shown in Figure 2. We use stable diffusion with text-to-image task as an example; similar procedures can be applied to other models.

### 3.1  GRADIENT COMPUTATION

Since the training dataset remains fixed after training, we can cache the loss gradient of each training data sample, as illustrated in Figure 2(a). For a given training pair $X^i = (p^i, x^i)$, and a timestep

$t$, the training data is processed through the diffusion model in the same way as during training, and a loss is computed between the model-predicted noise and a Gaussian noise $\epsilon$, where $\epsilon \sim \mathcal{N}(0, I)$. Back-propagation is then performed to obtain the gradient $g_t^i$ for the training data pair $X^i$ at timestep $t$. Once all gradients $\{g_1^i, g_2^i, \cdots, g_T^i\}$ for $X^i$ at all timesteps are obtained, we apply a compression technique to these gradients and cache the compressed versions for influence estimation. Furthermore, for tasks where only the top-$k$ most influential samples are required, we can construct a KNN index on the compressed gradients to enable efficient querying.

**Forward and Backward Passes.** In forwarding, following the same process as training, for a training pair $(p^i, x^i)$ and a timestep $t$, the prompt $p^i$ is passed through the encoder to obtain a prompt embedding, while the image $x^i$ is passed through a VAE to obtain a latent representation $z_0^i$. Gaussian noise is then generated from $\epsilon \sim \mathcal{N}(0, I)$ and added to the latent representation to obtain a noisy latent presentation. Then input the timestep $t$, noisy latent presentation $z_t^i = z_0^i + \epsilon$, and the embedding $z_p^i$ are then fed into the model for the forward pass.

After the forward pass, a loss is computed between the Gaussian noise $\epsilon$ and the predicted noise $\hat{\epsilon}$. Subsequently, back-propagation is performed to calculate the gradients for each parameter that requires a gradient. It is important to note that for models with adapters as illustrated in Figure 2(b), only the parameters associated with the adapters require gradients. After obtaining the gradients, we concatenate all of them and flatten them into a single vector. For a diffusion model with 2B parameters, this resulting gradient vector will have a length of 2B.

The number of training timesteps is typically 1,000, depending on the model training configuration. For a single training data sample, using a diffusion model with 2B parameters as an example, computing gradients for all $1,000$ timesteps is computationally intensive and costly, requiring over $7,450$ GB of storage. To mitigate this, similar to the inference process in diffusion models, we can sample a subset of timesteps from $t \in \{1, 2, \cdots, T\}$ instead of computing gradients for all timesteps, substantially reducing the computational and storage burden.

**Gradient Compression.** However, even storing the gradient vector for a single training data sample at just one timestep requires approximately 7 GB of storage. This becomes impractical for extremely large training datasets containing millions of samples. Therefore, gradient compression techniques are essential to enable caching gradients at this scale efficiently.

As previously discussed, some prior studies employ random projection to compress gradient vectors. However, for a diffusion model with 2B parameters, such compression requires a projection matrix of size 2B $\times v$, where $v$ is the dimension after compression. Even with a modest $v = 4096$, this matrix would require over 29 TB of storage. This makes these approaches feasible only for small models or LoRA-tuned models, substantially limiting their scalability.

Inspired by the prior works on vector compression (Li & Li, 2023; Lin et al., 2024), we compress the gradient vector through four steps: (1) padding, (2) permutation, (3) random projection, and (4) group addition. In the gradient compression process, we first pad the gradient vector to the smallest length $L_{\text{pad}}$ that can be evenly divided by $v$. Padding can be achieved by appending 0s to the original gradient vector until the desired length is reached. Next, we permute the gradient vector using a random permutation to disrupt any inherent structure in the vector representation. We then perform an element-wise multiplication of the permuted gradient vector with a random projection vector. The random projection vector is of the same length as the gradient vector and consists of elements randomly set to either -1 or 1 with equal probability. This step projects the gradient onto a randomized basis, reducing redundancy while preserving essential information. Finally, we divide the $L_{\text{pad}}$ elements of the gradient vector into $\frac{L_{\text{pad}}}{v}$ groups, summing up the elements within each group to produce the compressed vector of dimension $v$.

With this compression, we only need to store two components: a permutation vector that records the indices of the permutation (4 bytes per element) and a binary projection vector (1 bit per element). As a result, the storage requirement is significantly reduced, occupying just 7.45 GB for the gradients plus an additional 238 MB for the projection vector. This reduction makes it feasible to store and cache the gradients for influence estimation at scale.

**Normalization.** Some prior studies have highlighted the inherent instability of gradients in deep learning (Lin et al., 2024; Basu et al., 2021; Epifano et al., 2023; Ghorbani et al., 2019) particularly in extremely large models. This instability arises from the potential for unusually large weights and gradients in the model. In our experiments, we encountered this issue: the magnitude of some gradi-

ent values is found to be extremely large. Such large gradient values can dominate the inner product, leading to incorrect results. To address this, we apply L2 normalization to the gradient vector before compression, which effectively mitigates the impact of unusually large gradient magnitudes. Consequently, Equation 6 can be reformulated as:

$$\mathcal{I}_\theta(X^s, X^i) = e\bar\eta \sum_{t=1}^T \frac{\nabla_\theta \mathcal{L}\big(f_\theta(z_p^i, z_t^i, t), \epsilon\big)}{\big\|\nabla_\theta \mathcal{L}\big(f_\theta(z_p^i, z_t^i, t), \epsilon\big)\big\|_2} \cdot \frac{\nabla_\theta \mathcal{L}\big(f_\theta(z_p^s, z_t^s, t), \epsilon\big)}{\big\|\nabla_\theta \mathcal{L}\big(f_\theta(z_p^s, z_t^s, t), \epsilon\big)\big\|_2}$$

**Index Construction for KNN.** To further enhance the scalability of `DMin`, we introduce KNN search for tasks requiring only the top-$k$ most influential samples. After gradient compression, as shown in Figure 2(d), we concatenate all the compressed gradients across timesteps to construct a KNN index, enabling efficient querying during influence estimation. This approach is well-suited for large datasets, allowing for the retrieval of the top-$k$ most influential samples on the fly.

### 3.2 INFLUENCE ESTIMATION

After caching the compressed gradients, for a given generated image and its corresponding prompt, we compute and compress the gradient in the same way as for the training data samples to obtain the compressed gradient for the given sample. For exact influence estimation, we calculate the inner product between the compressed gradient of the given sample and the cached compressed gradients of each training sample across timesteps to obtain the influence scores. For KNN retrieval, we concatenate the compressed gradients across timesteps to query the KNN index and identify the top-$k$ most relevant training samples efficiently.

## 4 EXPERIMENTS

In this section, we present our experiments conducted on various models and settings to validate the effectiveness and efficiency of the proposed `DMin`.

Table 1: Sub-datasets used in experimental evaluation. (Full dataset is listed in Appendix C.2.)

| Subset | # Train | % of Training Data | # Test |
|---|---|---|---|
| Flowers | 162 | 1.74% | 34 |
| Lego Sets | 40 | 0.43% | 21 |
| Magic Cards | 1541 | 16.59% | 375 |

**Datasets.** For the conditional diffusion model, we combine six datasets from Huggingface and randomly select 80% of the data samples as the training dataset, resulting in 9,288 pairs of images and prompts. Due to page limitations, we list three datasets used for evaluation in Table 1: (1) Flowers, which includes 162 training pairs of flower images and corresponding descriptive prompts in our experiments, accounting for only 1.74% of the training dataset. (2) Lego Sets: This subset consists of 40 training pairs, where each image represents a Lego box accompanied by a description of the box, accounting for only 0.43% of the training dataset. (3) Magic Cards, which contains magic card images from scryfall with captions generated by Fuyu-8B (Bavishi et al., 2023) and BLIP (Li et al., 2022). For unconditional diffusion models, we mainly focus on MNIST and CIFAR-10. We include a detailed explanation of datasets in Appendix C.2.

**Models.** For conditional text-to-image diffusion model, we use three different models: (1) SD 1.4 with LoRA, (2) SD 3 Medium with LoRA and (3) SD 3 Medium (Full parameters). For unconditional diffusion model, we conduct experiments on two Denoising Diffusion Probabilistic Models (DDPM) trained on MNIST and CIFAR-10. The detailed settings of models are included in Appendix C.1. We fine-tune models on the combined training dataset mentioned above and evaluate them on the testing dataset. During gradient collection, we collect only the gradients of the parameters in the LoRA components for the LoRA-tuned model, whereas for the fully fine-tuned model, we collect the gradients of all parameters (Figure 2(b)).

**Baselines** We compare the proposed `DMin` against the following baselines: (1) Random Selection: Assigns an influence score to each training sample randomly. (2) SSIM (Brunet et al., 2012) Structural Similarity Index Measure (SSIM) between the training image and the generated image. (3) CLIP Similarity (Radford et al., 2021): Cosine similarity of embeddings computed by CLIP between the training image and the generated image. (4) LiSSA (Agarwal et al., 2017): A second-order influence estimation method that uses an iterative approach to compute the inverse Hessian-vector product. (5) DataInf (Kwon et al., 2024): An influence estimation method based on a closed-form expression for computational efficiency. We also evaluate a variant of DataInf where the Hessian

Table 2: Average detection rates of top-k most influential training data samples. Detection rate $=$ $\frac{\text{\# Samples from Same Subset among Top-k Training Samples}}{k}$ where $k = \{5, 10, 50, 100\}$, indicating the average proportion of samples from the same subset appearing in the top-k influential samples. "Ours (w/o Comp.)" indicates that the gradient vectors are not compressed, while "w/o Norm." signifies that the gradient vectors are not normalized. "Excatly" denotes exact inner product computation. The results for LiSSA, DataInf and D-TRAK on SD3 Medium (Full) are omitted due to hundreds of TB of cache. Moreover, at this scale, it is impractical for LiSSA and DataInf to approximate the Hessian inversion and for D-TRAK to compute a large random projection matrix.

| Model | Method | Flowers | | | | Lego Sets | | | | Magic Cards | | | |
|---|---|---|---|---|---|---|---|---|---|---|---|---|---|
| | | Top 5 | Top 10 | Top 50 | Top 100 | Top 5 | Top 10 | Top 50 | Top 100 | Top 5 | Top 10 | Top 50 | Top 100 |
| SD 1.4 (LoRA) | Random Selection | 0.0000 | 0.0000 | 0.0200 | 0.0100 | 0.0000 | 0.0000 | 0.0000 | 0.0000 | 0.2000 | 0.2000 | 0.0800 | 0.1300 |
| | SSIM | 0.2000 | 0.1000 | 0.0220 | 0.0130 | 0.0400 | 0.0400 | 0.0340 | 0.0240 | 0.2800 | 0.3500 | 0.4480 | 0.4290 |
| | CLIP Similarity | 0.0000 | 0.0000 | 0.0000 | 0.0000 | 0.0000 | 0.0000 | 0.0000 | 0.0000 | 0.4444 | 0.4005 | 0.3565 | 0.3830 |
| | LiSSA | 0.5143 | 0.4571 | 0.3486 | 0.2929 | 0.0000 | 0.0000 | 0.0040 | 0.0080 | 0.9667 | 0.9500 | 0.9600 | 0.9483 |
| | DataInf (Identity) | 0.4125 | 0.4062 | 0.3188 | 0.2687 | 0.0000 | 0.0000 | 0.0067 | 0.0100 | 0.9667 | 0.9500 | 0.9600 | 0.9483 |
| | DataInf (Hessian Inversion) | 0.4125 | 0.4062 | 0.3188 | 0.2687 | 0.0000 | 0.0000 | 0.0067 | 0.0100 | 0.9667 | 0.9500 | 0.9600 | 0.9483 |
| | Ours (w/o Comp. & Norm.) | 0.1333 | 0.1154 | 0.1138 | 0.1028 | 0.0000 | 0.0000 | 0.0047 | 0.0065 | 0.9637 | 0.9585 | 0.9402 | 0.9280 |
| | Ours (w/o Comp.) | **0.8872** | **0.8359** | **0.5836** | **0.3969** | 0.5647 | 0.4412 | 0.1435 | **0.0894** | **0.9778** | 0.9778 | 0.9911 | 0.9933 |
| | Ours ($v = 2^{12}$, Exactly) | 0.8667 | 0.8154 | 0.5713 | 0.3836 | 0.5176 | 0.3882 | 0.1435 | 0.0865 | **0.9778** | **0.9889** | **0.9933** | **0.9944** |
| | Ours ($v = 2^{16}$, Exactly) | 0.8615 | 0.8231 | 0.5718 | 0.3813 | 0.5529 | 0.4353 | **0.1447** | **0.0894** | **0.9778** | 0.9778 | 0.9911 | 0.9933 |
| | Ours ($v = 2^{20}$, Exactly) | 0.8667 | 0.8154 | 0.5713 | 0.3836 | **0.5647** | **0.4412** | 0.1435 | **0.0894** | **0.9778** | 0.9778 | 0.9911 | 0.9933 |
| | Ours ($v = 2^{12}$, KNN) | 0.8615 | 0.8128 | 0.5405 | 0.3585 | 0.5059 | 0.3647 | 0.1365 | 0.0824 | **0.9778** | **0.9889** | **0.9933** | **0.9944** |
| | Ours ($v = 2^{16}$, KNN) | 0.8615 | 0.8231 | 0.5723 | 0.3808 | 0.5412 | 0.4176 | 0.1388 | 0.0847 | **0.9778** | **0.9889** | 0.9889 | **0.9944** |
| SD 3 Medium (LoRA) | Random Selection | 0.0000 | 0.0000 | 0.0200 | 0.0100 | 0.0000 | 0.0000 | 0.0000 | 0.0000 | 0.2000 | 0.2000 | 0.0800 | 0.1300 |
| | SSIM | 0.1800 | 0.0900 | 0.0200 | 0.0160 | 0.0000 | 0.0000 | 0.0160 | 0.0190 | 0.0000 | 0.0067 | 0.0180 | 0.0347 |
| | CLIP Similarity | 0.0000 | 0.0000 | 0.0000 | 0.0000 | 0.0000 | 0.0000 | 0.0000 | 0.0000 | 0.0352 | 0.0363 | 0.0421 | 0.0438 |
| | LiSSA | 0.8889 | 0.8889 | 0.8622 | 0.8222 | 0.1111 | 0.1111 | 0.1244 | 0.1044 | 0.9091 | 0.9091 | 0.9091 | 0.9082 |
| | DataInf (Identity) | 0.8556 | 0.8556 | 0.7878 | 0.6683 | 0.1647 | 0.1176 | 0.0576 | 0.0424 | 0.8833 | 0.8917 | 0.8900 | 0.8883 |
| | DataInf (Hessian Inversion) | 0.8556 | 0.8556 | 0.7878 | 0.6683 | 0.1647 | 0.1176 | 0.0576 | 0.0424 | 0.8833 | 0.8917 | 0.8900 | 0.8883 |
| | Ours (w/o Comp. & Norm.) | 0.8974 | 0.8769 | 0.8010 | 0.6738 | 0.2588 | 0.1765 | 0.1024 | 0.0765 | 0.7935 | 0.7951 | 0.7965 | 0.7965 |
| | Ours (w/o Comp.) | **0.9128** | 0.8974 | 0.8390 | 0.7605 | 0.6118 | 0.5059 | 0.2318 | 0.1488 | **1.0000** | **1.0000** | **1.0000** | 0.9700 |
| | Ours ($v = 2^{12}$, Exactly) | 0.8974 | 0.8846 | 0.8318 | 0.7608 | 0.6000 | 0.5235 | 0.2306 | 0.1529 | 0.9837 | 0.9835 | 0.9751 | 0.9703 |
| | Ours ($v = 2^{16}$, Exactly) | 0.9077 | 0.8872 | 0.8405 | 0.7659 | 0.5765 | 0.5118 | 0.2224 | 0.1482 | 0.9848 | 0.9840 | 0.9761 | 0.9718 |
| | Ours ($v = 2^{20}$, Exactly) | 0.9077 | 0.8872 | 0.8385 | 0.7651 | 0.6000 | 0.5235 | 0.2294 | 0.1506 | 0.9848 | 0.9840 | 0.9762 | 0.9720 |
| | Ours ($v = 2^{12}$, KNN) | 0.9026 | 0.8949 | 0.8415 | 0.7641 | **0.7294** | **0.6529** | **0.3094** | **0.1924** | 0.9854 | 0.9851 | 0.9771 | 0.9717 |
| | Ours ($v = 2^{16}$, KNN) | **0.9128** | **0.9051** | **0.8472** | **0.7721** | 0.7059 | 0.6353 | 0.3035 | 0.1871 | 0.9864 | 0.9862 | 0.9785 | **0.9736** |
| SD 3 Medium (Full) | Random Selection | 0.0000 | 0.0000 | 0.0200 | 0.0100 | 0.0000 | 0.0000 | 0.0000 | 0.0000 | 0.2000 | 0.2000 | 0.0800 | 0.1300 |
| | SSIM | 0.1800 | 0.0967 | 0.0200 | 0.0117 | 0.0235 | 0.0176 | 0.0282 | 0.0206 | 0.0000 | 0.0000 | 0.0020 | 0.0160 |
| | CLIP Similarity | 0.0000 | 0.0000 | 0.0000 | 0.0000 | 0.0000 | 0.0000 | 0.0000 | 0.0000 | 0.2938 | 0.3412 | 0.4583 | 0.4982 |
| | Ours ($v = 2^{12}$, Exactly) | 0.9487 | 0.9000 | 0.5385 | 0.3567 | 0.5529 | 0.4412 | 0.1906 | 0.1165 | 0.9882 | 0.9882 | 0.9420 | 0.9063 |
| | Ours ($v = 2^{16}$, Exactly) | 0.9590 | 0.9308 | 0.5564 | 0.3690 | 0.5765 | 0.4765 | 0.2047 | 0.1282 | **0.9961** | 0.9902 | 0.9514 | 0.9220 |
| | Ours ($v = 2^{20}$, Exactly) | **0.9641** | **0.9333** | 0.5590 | **0.3708** | 0.5647 | 0.4765 | 0.2071 | 0.1306 | 0.9922 | 0.9922 | 0.9498 | 0.9202 |
| | Ours ($v = 2^{12}$, KNN) | 0.9282 | 0.8641 | 0.5354 | 0.3518 | 0.6125 | 0.5062 | 0.2025 | 0.1288 | 0.9880 | 0.9820 | 0.9472 | 0.9046 |
| | Ours ($v = 2^{16}$, KNN) | 0.9622 | 0.9108 | **0.5622** | 0.3695 | **0.6250** | **0.5437** | **0.2213** | **0.1419** | 0.9960 | **0.9960** | **0.9640** | **0.9308** |

Inversion matrix is replaced with an identity matrix. (6) D-TRAK (Ogueji et al., 2022): A first-order influence estimation method extended from TRAK (Park et al., 2023). (7) Journey-TRAK (Georgiev et al., 2023): An estimation method focusing on the sampling path in diffusion models. For the proposed `DMin`, we evaluate `DMin` under different scenarios, including exact estimation of influence scores for each training sample and KNN-based approximate searches for the top-$k$ most influential samples. Additionally, we experiment with varying compression levels: no compression, and $v = \{2^{12}, 2^{16}, 2^{20}\}$. The detailed information of baselines are reported in Appendix D.1.

**KNN.** We utilized hierarchical navigable small world (HNSW) algorithm (Malkov & Yashunin, 2020) for KNN in our experiment, and provide the results of ablation study in Appendix D.

## 4.1 PERFORMANCE ON CONDITIONAL DIFFUSION MODELS

The goal of this experiment is to confirm the effectiveness of different methods in identifying influential training samples within the training dataset.

**Visualization.** Figure 1 illustrates several examples, showing the generated image and its corresponding prompt in the first column, followed by the training samples ranked from highest to lowest influence, arranged from left to right. These examples demonstrate that the proposed `DMin` method successfully retrieves training image samples with content similar to the generated image and prompt. Additional visualizations are provided in Appendix E.

**Qualitative Analysis.** Unlike prior studies focusing on small diffusion models, the diffusion models used in our experiments are substantially larger, making it impractical to retrain them for leave-one-out evaluation. Consequently, we assess the detection rate in our experiments, as shown in Table 2, which reflects the average proportion of similar content from the training dataset retrieved by the top-$k$ most influential samples.

**Datasets.** As mentioned earlier, our training dataset is a combination of six datasets. As shown in Table 1, we report evaluations on three subsets: Flowers, Lego Sets, and Magic Cards, as these subsets are more distinct from the others. For example, given a prompt asking the model to generate a magic card, the generated image should be more closely related to the Magic Cards subset rather than the Flowers or Lego Sets subsets, as the knowledge required to generate magic cards primarily

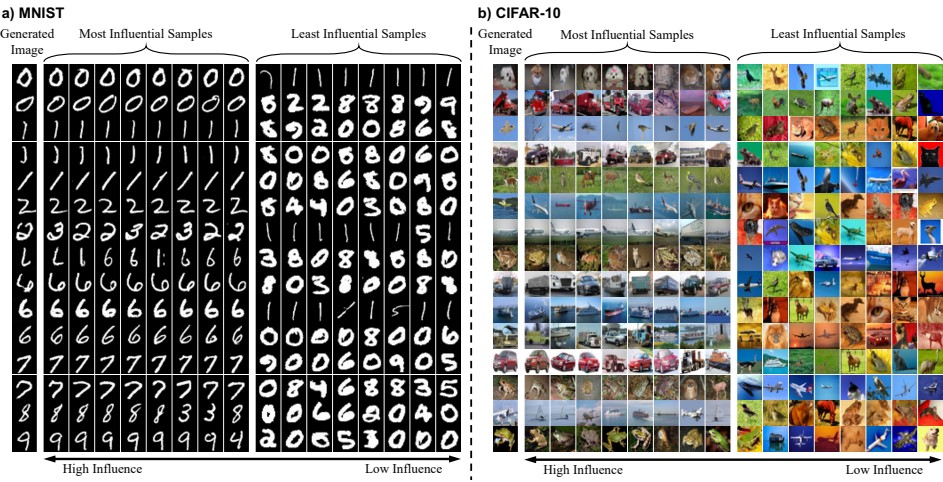

Figure 3: Examples of generated images alongside the most and least influential samples (from left to right) as estimated by DMin for unconditional DDPM models on the MNIST and CIFAR-10.

Table 3: Storage requirements for caching per-sample and dataset gradients (9,288 samples), comparing compressed and uncompressed methods across models. The table shows storage and compression ratios of our method across levels, with LiSSA and DataInf storing gradients uncompressed.

| Method | SD 1.4 (LoRA, 10 Timesteps) | | | SD 3 Medium (LoRA, 10 Timesteps) | | | SD 3 Medium (Full, 5 Timesteps) | | |
|---|---|---|---|---|---|---|---|---|---|
| | Size (Per Sample) | Size (Training Dataset) | Compression Ratio | Size (Per Sample) | Size (Training Dataset) | Compression Ratio | Size (Per Sample) | Size (Training Dataset) | Compression Ratio |
| Gradient w/o Comp. | 30.41 MB | 275.82 GB | 100% | 45 MB | 408.16 GB | 100% | 37.42 GB | 339.39 TB | 100% |
| Ours ($v = 2^{12}$) | 160 KB | 1.45 GB | 0.53% | 160 KB | 1.45 GB | 0.36% | 80 KB | 726 MB | 0.00017% |
| Ours ($v = 2^{16}$) | 2.5 MB | 22.68 GB | 8.22% | 2.5 MB | 22.68 GB | 5.56% | 1.25 MB | 11.34 GB | 0.0028% |
| Ours ($v = 2^{20}$) | - | - | - | - | - | - | 20 GB | 181.41 GB | 0.044% |

originates from the Magic Cards subset. Similarly, the knowledge for generating images containing Lego comes predominantly from the Lego Sets subset. Therefore, for a prompt belonging to one of the test subsets – Flowers, Lego Sets, or Magic Cards – the most influential training samples are highly likely to originate from the same subset. This implies that a number of training samples from the corresponding subset should be identified among the top-$k$ most influential samples.

We begin by generating images using the prompts from the test set of each subset – Flowers, Lego Sets, and Magic Cards. For each test prompt and its generated image, we estimate the influence score for every training data sample and select the top-$k$ most influential training samples with the highest influence score. We then calculate the detection rate as Detection Rate $=$ $\frac{\text{\# Samples from Same Subset among top-}k\text{ Training Samples}}{k}$.

**Results.** We report the average detection rate for each test set of subsets in Table 2. Compared to the baselines, our proposed DMin obtains the best performance on all subsets. Compared to the baselines, our proposed DMin achieves the best performance across all subsets. The detection rates for top-50 and top-100 on Lego Sets are lower because the Lego Sets training dataset contains only 40 samples (0.43% of the total). Across all subsets and different values of $k$, $v = 2^{16}$ achieves the best performance in most cases, whether using KNN or exact inner product computation. Additionally, compared to our method without compression, removing normalization substantially decreases performance, confirming that normalization mitigates the instability of gradients in extremely deep models. Interestingly, KNN search often outperforms exact inner product computation in our experiments across all models and subsets. This improvement is likely due to KNN's ability to approximate the search process, capturing a broader and more representative subset of neighbors.

## 4.2 TIME AND MEMORY COST

The computational cost of both time and memory is critical for evaluating the scalability of influence estimation methods, especially when applied to large diffusion models.

**Time.** Table 4 demonstrates the time consumption on estimate the influence score for every training sample in the training dataset for a single test sample. The gradient computation and caching times for two LoRA-tuned models are nearly identical due to the small model size across different methods: (1) SD 1.4 (LoRA): around 8 GPU hours, (2) SD 3 Medium (LoRA): around 24 GPU

Table 4: Time cost comparison and speedup relative to our method without compression. Time cost refers to the time to estimate influence scores for all training samples per test sample (in seconds).

| Method | SD 1.4 (LoRA) | | SD 3 Medium (LoRA) | | SD 3 Medium (Full) |
|---|---|---|---|---|---|
| | Time (seconds/test sample) | Speedup (vs. w/o Comp.) | Time (seconds/test sample) | Speedup (vs. w/o Comp.) | Time (seconds/test sample) |
| LiSSA | 2,939.283 | 0.02x | 2,136.701 | 0.19x | - |
| DataInf (Identity) | 206.385 | 0.34x | 201.923 | 2.02x | - |
| DataInf (Hessian Inversion) | 1,187.841 | 0.06x | 932.762 | 0.44x | - |
| D-TRAK | 345.223 | 0.20x | 833.850 | 0.49x | - |
| Ours (w/o Comp.) | 70.590 | 1x | 407.511 | 1x | - |
| Ours ($v = 2^{12}$, Exact) | 8.193 | 8.62x | 14.238 | 28.62x | 9.866 |
| Ours ($v = 2^{16}$, Exact) | 41.026 | 1.72x | 135.462 | 3.01x | 18.900 |
| Ours ($v = 2^{20}$, Exact) | 99.307 | 0.71x | 623.610 | 0.65x | 100.880 |
| Ours ($v = 2^{12}$, KNN, Top-5) | **0.004** | **18,100.51x** | **0.004** | **101,877.75x** | **0.009** |
| Ours ($v = 2^{12}$, KNN, Top-50) | 0.018 | 3,921.78x | 0.010 | 40,751.10x | 0.014 |
| Ours ($v = 2^{12}$, KNN, Top-100) | 0.033 | 2,139.15x | 0.019 | 21,447.95x | 0.131 |
| Ours ($v = 2^{16}$, KNN, Top-5) | 0.073 | 967.01x | 0.065 | 6,269.40x | 0.097 |
| Ours ($v = 2^{16}$, KNN, Top-50) | 0.393 | 179.62x | 0.227 | 1,792.04x | 0.485 |
| Ours ($v = 2^{16}$, KNN, Top-100) | 0.736 | 95.91x | 0.406 | 1,003.72x | 0.784 |

hours, and (3) SD 3 Medium (full): 330 GPU hours. Additionally, the index construction process only takes a few minutes. Our proposed methods demonstrate substantial efficiency improvements, particularly with KNN search. For instance, on the smallest subset—Lego Sets, which contains only 21 test samples—estimating the influence score for the entire training dataset takes 17 hours with LiSSA, 7 hours with DataInf (Hessian Inversion), and 2 hours with D-TRAK. In contrast, our method with $v = 2^{12}$ and $k = 5$ requires only 0.084 seconds, and even for $k = 100$, it takes only 0.69 seconds while achieving the best performance.

**Memory.** Table 3 compares the storage requirements for caching per-sample gradients and the entire training dataset (9,288 samples) across different models, with and without compression. Without compression, gradient storage is substantially large, reaching 339.39 TB for SD 3 Medium (Full). In contrast, our method achieves drastic reductions in storage size with various compression levels. For example, using $v = 2^{12}$, the storage for SD 3 Medium (Full) is reduced to just 726 MB, achieving a compression ratio of 0.00017%, demonstrating the scalability and efficiency of our approach for handling large-scale models.

### 4.3 Unconditional Diffusion Models

We evaluate the performance of the proposed `DMin` on unconditional diffusion models using DDPM on the MNIST and CIFAR-10 datasets. Figure 3 illustrates examples of generated images and the corresponding most and least influential training samples as identified by our method. On MNIST (Figure 3(a)), the most influential samples for each generated digit closely resemble the generated image, validating the effectiveness of our approach. Similarly, for CIFAR-10 (Figure 3(b)), our method retrieves relevant training samples with similar content. These results highlight the scalability and reliability of our method for detecting influential samples in unconditional diffusion models.

Table 5 reports the detection rate compared to baseline methods Journey-TRAK and D-TRAK. Our method consistently outperforms both baselines across all metrics, achieving substantially higher detection rates. For instance, with $v = 2^{16}$, our method achieves a detection rate of 0.8006 for Top-5 on MNIST, while Journey-TRAK and D-TRAK achieve only 0.2560 and 0.1264, respectively.

Table 5: Detection Rate compared with Journey-TRAK and D-TRAK for the unconditional DM (DDPM) on MNIST.

| Method | Top 5 | Top 10 | Top 50 | Top 100 |
|---|---|---|---|---|
| Journey-TRAK | 0.2560 | 0.2190 | 0.1732 | 0.1513 |
| D-TRAK | 0.1264 | 0.1410 | 0.1382 | 0.1272 |
| Ours ($v = 2^{12}$, Exact) | 0.4376 | 0.4315 | 0.4094 | 0.4027 |
| Ours ($v = 2^{16}$, Exact) | 0.8006 | 0.7901 | 0.7408 | 0.7098 |

## 5 Conclusion

In this paper, we introduce `DMin`, a scalable framework for estimating the influence of training data samples on images generated by diffusion models. The proposed `DMin` scales effectively to diffusion models with billions of paramters by substantially reducing storage requirements from hundreds of TBs to MBs or KBs for SD 3 Medium with full parameters. Additionally, `DMin` can retrieve the top-$k$ most influential training sample in 1 second by KNN, demonstrating the scalability of the proposed `DMin`. Our empirical results further confirm `DMin`'s effectiveness and efficiency.

## ETHICS STATEMENT

All authors have read and will adhere to the ICLR Code of Ethics. This work studies influence estimation for diffusion models without involving human subjects or personally identifiable information. We use only publicly available datasets (e.g., MNIST, CIFAR-10, and image–text datasets from Hugging Face) and model checkpoints, respecting their licenses and terms of use; for third-party assets we provide pointers or scripts rather than redistributing restricted content. Our experiments fine-tune models and synthesize images programmatically; we took care to avoid harmful or offensive content in examples. Because influence estimation could be misinterpreted as certifying provenance, we emphasize that our scores are diagnostic signals, not legal attribution.

## REPRODUCIBILITY STATEMENT

We provide a code repository with complete evaluation code, configuration files. Model versions and training/fine-tuning settings are documented, as are software/hardware details; random seeds are fixed where applicable, and nondeterministic components are noted. Upon publication, we will open-source both the codebase and the dataset artifacts after acceptance.

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

## A  THE USE OF LLMs

Large Language Models (LLMs) were employed exclusively to refine the writing of this paper by improving grammar, clarity, and readability. They were not used for research conception, experimental design, implementation, or analysis. The authors assume full responsibility for all content.

## B  RELATED WORK

Influence estimation has been a critical area of research in understanding the impact of individual training samples on machine learning models (Schioppa et al., 2022; Park et al., 2023; Yang et al., 2024b; Chhabra et al., 2024). Early work by Koh & Liang (2017); Agarwal et al. (2017) proposed second-order Hessian-based methods to approximate the effect of a training sample. However, approximating a Hessian inversion becomes computationally prohibitive for large-scale datasets and modern models containing billions of parameters. To address this issue, some studies proposed first order approaches for influence estimation (Pruthi et al., 2020; Park et al., 2023). However, even with first-order methods, scaling to large datasets still encounter storage challenges. For example, storing the gradient of a 2B diffusion model for 10,000 data samples across 10 timesteps requires over 700 TB of storage.

To reduce the storage and computational demands, some studies leverage dimension reduction techniques (Park et al., 2023; Ogueji et al., 2022; Georgiev et al., 2023; Hammoudeh & Lowd, 2024), such as random projection. However, while random projection can substantially reduce the dimension of gradient vector, the projection matrix itself becomes a scalability bottleneck in large models. For instance, in a model with 2B parameters, a projection matrix mapping gradients to a compressed dimension of 32,768 would require over 500 GB of storage. These constraints highlight the need for more efficient and scalable approaches.

## C  EXPERIMENTAL SETTINGS.

In this section, we report the detailed setting and environments for our experiments.

**Implementation Details.** We provide an open-source PyTorch implementation with multiprocessing support[2]. We leverage Huggingface, Accelerate, Transformers, Diffusers and Peft in our implementation.

**Experimental Environments.** Our experiments are conducted on four different types of servers: (1) Servers running Red Hat Enterprise Linux 7.8, equipped with Intel(R) Xeon(R) Platinum 8358 processors (2.60GHz) with 32 cores, 64 threads, 4 A100 80G GPUs, and 1TB of memory. (3) Servers running Red Hat Enterprise Linux 7.8, containing Intel(R) Xeon(R) Gold 6226R CPUs @ 2.90GHz with 16 cores, 32 threads, 2 A100 40G GPUs, and 754GB of memory. (4) A server running Ubuntu 20.04.6 LTS, featuring 2 H100 GPUs, dual Intel(R) Xeon(R) Gold 6438N processors (3.60GHz) with 32 cores, 64 threads, and 1.48TB of memory. To ensure a fair comparison, all experiments measuring time cost and memory consumption are conducted on server 1, while other experiments are distributed across the different server types.

### C.1  MODELS

This study evaluates the performance of the following models: (1) SD 1.4 with LoRA: This model integrates Stable Diffusion 1.4 (SD 1.4) with Low-Rank Adaptation (LoRA), a technique that fine-tunes large models efficiently by adapting specific layers to the target task while maintaining most of the original model's structure. (2) SD 3 Medium with LoRA: Utilizing the Stable Diffusion 3 Medium (SD 3 Medium) base model, this configuration applies LoRA for task-specific adaptation. The medium-sized architecture of SD 3 balances computational efficiency with high-quality generation performance. (3) SD 3 Medium: A standalone version of Stable Diffusion 3 Medium, serving as a baseline for comparison against the LoRA-enhanced models. This version operates without any additional fine-tuning, showcasing the model's capabilities in its default state. Additionally, we include the hyperparameter settings in Table 7.

### C.2  DATASETS

In this section, we introduce the datasets used on our experiments of conditional diffusion models and unconditional diffusion models.

**Dataset Combination.** For conditional diffusion models, We combine six datasets from Huggingface: (1) magic-card-captions by clint-greene, (2) midjourney-detailed-prompts by Mohame-

---

[2] https://anonymous.4open.science/r/DMin

dRashad, (3) diffusiondb-2m-first-5k-canny by HighCWu, (4) lego-sets-latest by merve, (5) pokemon-blip-captions-en-ja by svjack, and (6) gesang-flowers by Albe-njupt. Additionally, we introduced noise to 5% of the data, selected randomly, and appended it to the dataset to enhance robustness. Finally, we split the data, allocating 80% (9,288 samples) for training and the remaining 20% for testing. For unconditional diffusion models, we use two classic datasets: (1) MNIST and (2) CIFAR-10.

**Dataset Examples.** Figure 4 showcases randomly selected examples from each dataset. For clarity, prompts are excluded from the visualizations. The original prompts can be accessed in the corresponding Huggingface datasets.

Table 6: Average detection rate on different $\text{ef}_{\text{construction}}$, $M$ and ef in HNSW implementation.

| ef | Subset | $M$ | $\text{ef}_{\text{construction}}$ | | | | | |
|----|--------|-----|-----|-----|-----|-----|-----|-----|
| | | | 50 | 100 | 200 | 300 | 400 | 500 |
| 200 | Flowers | 4 | 0.8405 | 0.8349 | 0.8405 | 0.8405 | 0.8405 | 0.8405 |
| | | 8 | 0.8410 | 0.8410 | 0.8415 | 0.8415 | 0.8415 | 0.8415 |
| | | 16 | 0.8410 | 0.8415 | 0.8415 | 0.8415 | 0.8415 | 0.8415 |
| | | 32 | 0.8410 | 0.8415 | 0.8415 | 0.8415 | 0.8415 | 0.8415 |
| | | 48 | 0.8410 | 0.8415 | 0.8415 | 0.8415 | 0.8415 | 0.8415 |
| | Lego Sets | 4 | 0.2800 | 0.3035 | 0.3094 | 0.3094 | 0.3082 | 0.3082 |
| | | 8 | 0.3082 | 0.3082 | 0.3094 | 0.3094 | 0.3094 | 0.3094 |
| | | 16 | 0.3082 | 0.3082 | 0.3094 | 0.3082 | 0.3094 | 0.3094 |
| | | 32 | 0.3082 | 0.3082 | 0.3094 | 0.3094 | 0.3094 | 0.3094 |
| | | 48 | 0.3082 | 0.3082 | 0.3094 | 0.3094 | 0.3094 | 0.3094 |
| | Magic Cards | 4 | 0.9770 | 0.9772 | 0.9772 | 0.9772 | 0.9772 | 0.9772 |
| | | 8 | 0.9772 | 0.9772 | 0.9772 | 0.9771 | 0.9771 | 0.9771 |
| | | 16 | 0.9771 | 0.9771 | 0.9771 | 0.9771 | 0.9771 | 0.9771 |
| | | 32 | 0.9771 | 0.9771 | 0.9771 | 0.9771 | 0.9771 | 0.9771 |
| | | 48 | 0.9771 | 0.9771 | 0.9771 | 0.9771 | 0.9771 | 0.9771 |
| 1000 | Flowers | 4 | 0.8415 | 0.8415 | 0.8415 | 0.8415 | 0.8415 | 0.8415 |
| | | 8 | 0.8415 | 0.8415 | 0.8415 | 0.8415 | 0.8415 | 0.8415 |
| | | 16 | 0.8415 | 0.8415 | 0.8415 | 0.8415 | 0.8415 | 0.8415 |
| | | 32 | 0.8415 | 0.8415 | 0.8415 | 0.8415 | 0.8415 | 0.8415 |
| | | 48 | 0.8415 | 0.8415 | 0.8415 | 0.8415 | 0.8415 | 0.8415 |
| | Lego Sets | 4 | 0.2847 | 0.3035 | 0.3094 | 0.3094 | 0.3094 | 0.3094 |
| | | 8 | 0.3094 | 0.3094 | 0.3094 | 0.3094 | 0.3094 | 0.3094 |
| | | 16 | 0.3094 | 0.3094 | 0.3094 | 0.3094 | 0.3094 | 0.3094 |
| | | 32 | 0.3094 | 0.3094 | 0.3094 | 0.3094 | 0.3094 | 0.3094 |
| | | 48 | 0.3094 | 0.3094 | 0.3094 | 0.3094 | 0.3094 | 0.3094 |
| | Magic Cards | 4 | 0.9770 | 0.9771 | 0.9771 | 0.9771 | 0.9771 | 0.9771 |
| | | 8 | 0.9771 | 0.9771 | 0.9771 | 0.9771 | 0.9771 | 0.9771 |
| | | 16 | 0.9771 | 0.9771 | 0.9771 | 0.9771 | 0.9771 | 0.9771 |
| | | 32 | 0.9771 | 0.9771 | 0.9771 | 0.9771 | 0.9771 | 0.9771 |
| | | 48 | 0.9771 | 0.9771 | 0.9771 | 0.9771 | 0.9771 | 0.9771 |

Table 7: Hyperparameter settings for model training.

| Method | Learning Rate | Batch Size | # Epochs | Image Size | LoRA Rank | LoRA Alpha | LoRA Target Layers | Precision |
|--------|---------------|------------|----------|------------|-----------|------------|--------------------|-----------|
| SD 1.4 (LoRA) | 0.001 | 64 | 150 | $512 \times 512$ | 4 | 8 | [to_k, to_q, to_v, to_out.0] | float32 |
| SD 3 Medium (LoRA) | 0.001 | 64 | 150 | $512 \times 512$ | 4 | 8 | [to_k, to_q, to_v, to_out.0] | float32 |
| SD 3 Medium (Full) | 0.0001 | 64 | 150 | $512 \times 512$ | - | - | - | float32 |

# D  ABLATION STUDY

To better understand the impact of key parameters on the performance of the HNSW implementation, we conducted an ablation study by varying the graph-related parameters $M$ and ef, as well as the construction parameter $\text{ef}_{\text{construction}}$. Table 6 summarizes the average detection rates across three subsets: Flowers, Lego Sets, and Magic Cards, under a range of settings on SD 3 Medium with LoRA ($v = 2^{12}$).

The parameter $M$ determines the maximum number of connections for each node in the graph. A larger $M$ leads to denser graphs, which can improve accuracy at the cost of increased memory and computational overhead. The parameter $\text{ef}_{\text{construction}}$ controls the size of the dynamic list of candidates during graph construction, influencing how exhaustive the neighborhood exploration is

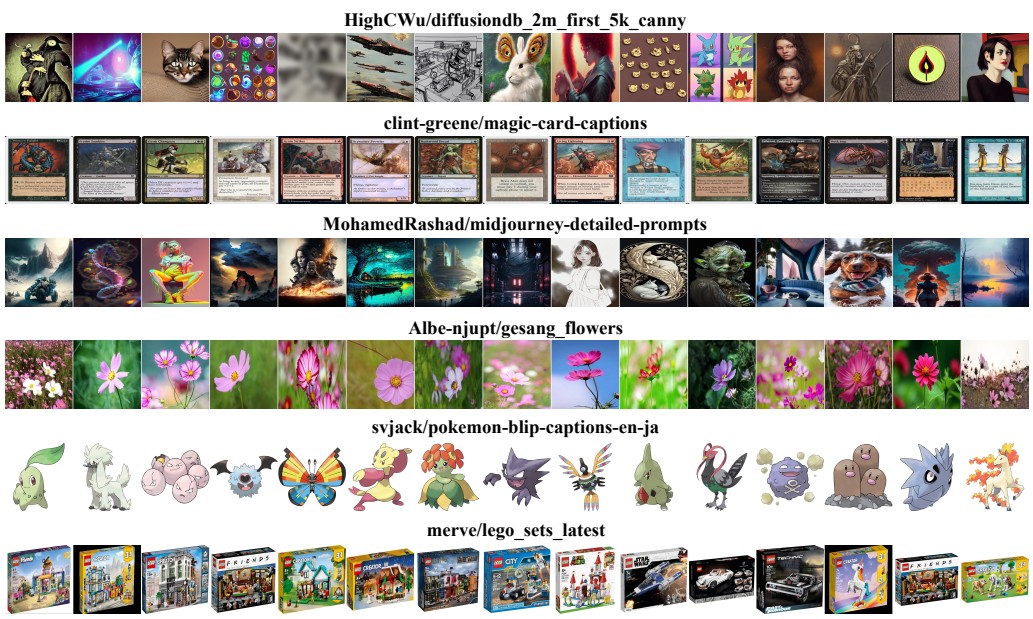

Figure 4: Examples of each dataset used in experiments.

during index creation. Lastly, the query-time parameter ef defines the size of the candidate list used during the search operation, directly affecting the trade-off between accuracy and efficiency.

Across the three datasets, the Magic Cards consistently exhibited high detection rates, exceeding 97.7% in all configurations, indicating that it is less sensitive to parameter tuning. In contrast, the Lego Sets showed significant variability. For ef $= 200$, the detection rate improved notably with higher values of $M$ (e.g., from 28% at $M = 4$ to 30.82% at $M = 8$ in ef $= 200$ and ef$_{construction}$ $= 50$), but beyond ef$_{construction}$ $= 100$, further increases in ef$_{construction}$ provided diminishing returns. This suggests that while denser graphs and more exhaustive index construction improve accuracy for complex datasets, the benefits plateau at a certain point. For the Flowers, the detection rates remained stable at approximately 84.1% across all parameter settings, indicating that this dataset is robust to variations in $M$ and $ef$.

### D.1 BASELINES

We compare the proposed `DMin` with seven baselines:

- **Random Selection:** Serves as a simple yet essential baseline where data points are selected randomly. This approach tests the performance against non-informed selection methods and ensures fairness in evaluation.
- **SSIM:** A widely-used metric for assessing the similarity between two images or signals. This baseline tests the performance of similarity measures rooted in visual or structural fidelity.
- **CLIP Similarity:** Exploits the feature embeddings generated by the CLIP, comparing their cosine similarity. It assesses how well general-purpose visual-language models can capture meaningful data relationships.
- **LiSSA:** Measures the influence of training points on the model's predictions by linearizing the loss function. This baseline provides a data-centric perspective on sample selection based on their impact on model training.
- **DataInf:** Employs data influence techniques to prioritize training samples that most strongly influence specific predictions. It represents methods that utilize influence diagnostics in data selection.
- **D-TRAK:** Focuses on tracking data's training impact using gradient information. This baseline evaluates approaches that harness gradient dynamics for data importance measurement.

- **Journey-TRAK:** Similar to D-TRAK but extends it to capture cumulative training effects over extended iterations. It benchmarks the ability of methods to consider long-term training trajectories in sample importance.

## E  SUPPLEMENTAL VISUALIZATION FOR CONDITIONAL DIFFUSION MODELS

We provide additional visualizations for unconditional models on the MNIST dataset in Figure 5 and for conditional models in Figure 6. Examples for other methods are omitted as they are nearly identical.

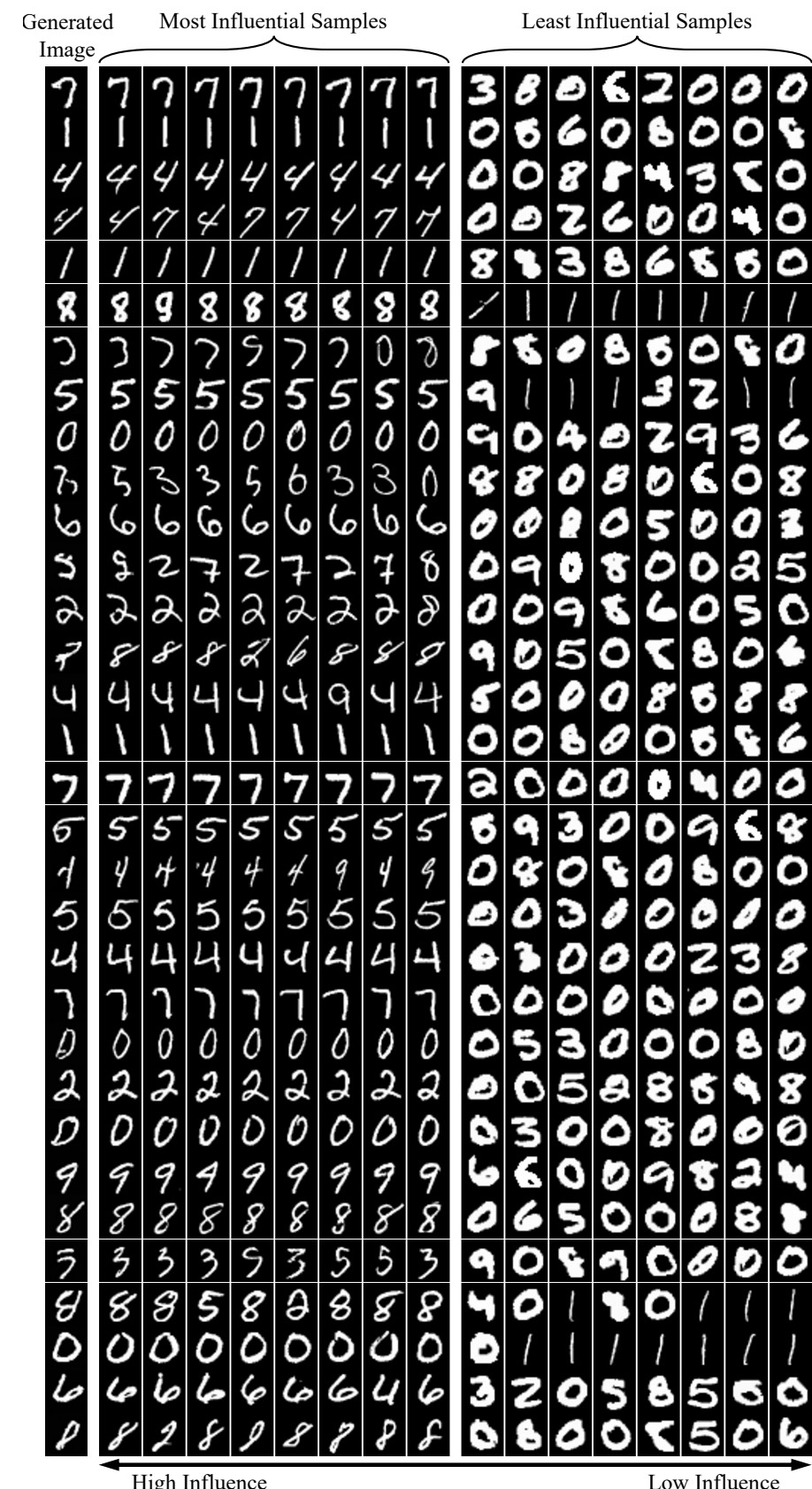

Figure 5: Additional visualization for unconditional diffusion model on the MNIST dataset.

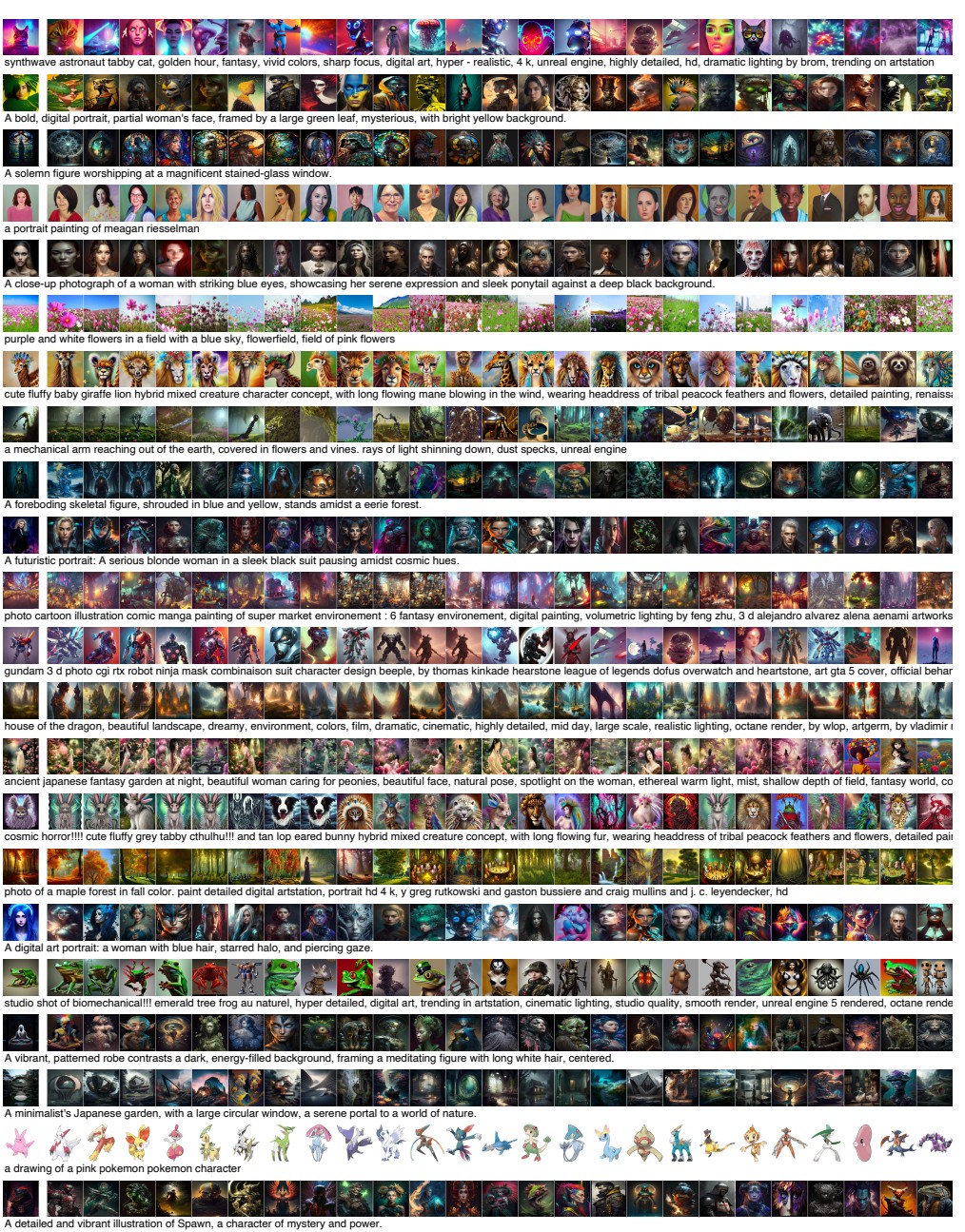

Figure 6: Examples of the top-25 most influential training data samples for the generated image (the 1-st column) on SD 3 Medium with LoRA, shown from high to low influence from left to right.

