# OpenReview forum: "DMin: Scalable Training Data Influence Estimation for Diffusion Models"
_ICLR.cc/2026/Conference — ICLR 2026 Conference Withdrawn Submission_

### Official Review · Reviewer_5RHZ · 2025-10-25

**Soundness:** 2
**Presentation:** 2
**Contribution:** 1
**Rating:** 4
**Confidence:** 5

**Summary:**

This paper proposes DMin, a framework for estimating the influence of individual training samples on generated images from diffusion models (DMs). The core idea is to compute and compress per-sample gradients across diffusion timesteps, then use inner products (or KNN search on compressed representations) to approximate influence scores. The method claims to scale to full-parameter billion-parameter diffusion models (e.g., SD3 Medium) by reducing storage from terabytes to kilobytes via a custom gradient compression pipeline involving permutation, sign random projection, and group summation. Experiments are conducted on small subsets of a combined Hugging Face dataset (e.g., “Flowers”, “Lego Sets”, “Magic Cards”) and standard unconditional benchmarks (MNIST, CIFAR-10), reporting “detection rates” as a proxy for influence estimation quality.

**Strengths:**

Scalability Focus: The paper correctly identifies a critical bottleneck in existing influence estimation methods for diffusion models, namely, prohibitive storage and computation costs for large models, and attempts to address it.

Engineering Efficiency: The proposed gradient compression technique (padding → permutation → sign projection → group sum) is simple and yields impressive storage reductions (e.g., from 339 TB to <1 GB for SD3 Medium).

Practical Tooling: The authors provide an open-source implementation with multiprocessing support, which could be useful for the community if the method is validated.

**Weaknesses:**

**1. Foundational Assumption Limits Validity**

The entire influence estimation formulation relies on a previous work (The Inverse-Hessian-based estimator for Influence Functions by Koh), implicitly assuming convexity of the loss landscape. This is a strong and unrealistic assumption for modern billion-parameter diffusion models trained with non-convex objectives and complex architectures (e.g., transformers). As a result, the theoretical grounding of DMin is fragile, and the computed “influence scores” may not reflect true causal or counterfactual effects. The paper does not discuss this limitation or provide ablation studies to validate the approximation quality under non-convexity.

**2. Miss Key Related Works**

The paper overlooks several recent, non-convex, scalable influence estimation methods that directly challenge its assumptions and claims, including:

[a]. Data Cleansing for Models Trained with SGD

[b]. Data Pruning via Moving-one-Sample-out

[c]. Z0-Inf: Zeroth Order Approximation for Data Influence”


**3. Lack of Convincing Evaluation**

All experiments are conducted on tiny, synthetic, or highly curated subsets (e.g., 40 Lego samples out of 9k total). These are essentially toy settings that do not reflect real-world data scale, diversity, or ambiguity. Crucially, there is no leave-one-out retraining or counterfactual generation to ground-truth influence, which is the gold standard in data attribution. No results are shown on large-scale, realistic datasets (e.g., LAION-5B, COYO), where influence estimation is most needed.

**Questions:**

See Weakness.

---

### Official Review · Reviewer_HdYt · 2025-10-29

**Soundness:** 3
**Presentation:** 2
**Contribution:** 2
**Rating:** 2
**Confidence:** 3

**Summary:**

This paper proposes DMin (Diffusion model influence), which incorporates several engineering optimizations on top of a TracIn kind first-order influence approximation, such as gradient compression, normalization, and approximate indexing. It is shown that these optimizations allow DMin to work with large diffusion models, such as stable-diffusion 3 medium.

**Strengths:**

1. The paper is a solid work, and the presentation is clear.
2. The experiment result seems promising, and it is indeed quite scalable as promised.

**Weaknesses:**

1. The core influence estimator appears to be a straightforward TracIn-style first-order gradient similarity, while the main additions are engineering choices (e.g., gradient compression, normalization, and approximate indexing) rather than a fundamentally new influence formulation.
2. I understand the difficulty of causal evaluation at the scale of modern diffusion models, but the current subset-detection metric primarily only captures similarity, which I don't believe is a good measure for data influence. Without any counterfactual analyses, it is hard to conclude that the method measures causal contribution.

**Questions:**

1. Although your method targets scalable attribution for large diffusion models, could you also report LDS result on a small diffusion model to quantify the accuracy trade-offs introduced by the memory-saving design choices?
2. Even though I raised concerns about the novelty and evaluation of the proposed approach, I agree that the problem of scalable attribution is important in practice. If the authors can demonstrate more evidence of the novelty and effectiveness of the proposed approach, I am happy to raise my score.

---

### Official Review · Reviewer_YDQC · 2025-10-29

**Soundness:** 2
**Presentation:** 3
**Contribution:** 2
**Rating:** 2
**Confidence:** 4

**Summary:**

This paper focuses on the problem of training data attribution for large-scale diffusion models. Prior work has been difficult to scale computationally due to the need to compute the inverse of a Hessian matrix, while first-order methods, though theoretically feasible, face extremely high storage bottlenecks.
This paper follows the framework of prior first-order attribution methods, simplifying the attribution problem to the dot product of two gradient vectors (the training sample gradient and the test sample gradient). Its main contribution lies in proposing an engineering-based gradient compression technique to solve the storage demands of previous methods. This method is a heuristic $O(D)$ compression scheme that forcibly compresses TB-level gradient caches down to MB or KB levels through (1) random permutation, (2) random projection (multiplying by $\pm 1$), and (3) group addition.
The authors then conducted extensive experiments to validate the advantages of this compression technique in computational efficiency and storage, and demonstrated its effectiveness in retrieving the top-k most influential samples.

**Strengths:**

1. Clear Structure and Readability: The paper is well-structured and clearly written, presenting a logical argument that is easy to follow.
2. Solves a Critical Engineering Bottleneck: The primary contribution is practical. It identifies and solves the key storage bottleneck (reducing requirements from TBs to KBs) that made prior first-order attribution methods infeasible, providing an engineering solution that enables attribution for large-scale models.
3. Extensive Large-Scale Validation: The approach is validated with extensive experiments on its target—large-scale diffusion models (e.g., Stable Diffusion). The results successfully demonstrate the method's efficiency, effectiveness, and scalability.

**Weaknesses:**

This paper suffers from significant flaws in its positioning, motivation, and experimental design.
1. Mis-Positioning of Contribution and Omission of Key Literature
The work is incorrectly positioned as a novel, standalone attribution method. In reality, its contribution is an engineering component—a compression module that could be applied as a plug-and-play accelerator to any gradient-based attribution method, including second-order (e.g., K-FAC, D-TRAK, DAS) and first-order (e.g., Diffusion-TracIn) approaches.
The paper validates its component using a first-order framework that is functionally identical to Diffusion-TracIn, yet it critically fails to discuss or cite this foundational work. Key strategies, such as the L2 normalization of gradients, were already introduced and discussed in Diffusion-TracIn but are presented here without proper attribution.
Furthermore, the paper fails to discuss overlapping contributions from DAS. A section in the DAS paper already discusses and validates available acceleration strategies, including gradient compression, selected timestep and normalization, which directly overlap with the central claims of this work. This omission creates a false sense of novelty.
2. Fundamentally Flawed Motivation and Lack of Theoretical Grounding
The paper's motivation for its novel compression method is based on a major factual error. The authors construct a strawman argument by claiming that prior work (like D-TRAK, line 75) requires storing a massive, dense Johnson-Lindenstrauss (JL) projection matrix (e.g., "a $2B \times 32,768$ matrix").
This premise is demonstrably false. Prior art (TRAK, D-TRAK, DAS) does not use dense JL; it employs Fast-JL techniques, which algorithmically compute the projection in $O(D \log D)$ time precisely to avoid storing such a matrix. The paper’s entire motivation, which is built on attacking this non-existent storage bottleneck, is therefore invalid.
Compounding this, the paper's own projection method (permutation, random signs, and group addition) is presented as a purely engineering heuristic. Unlike Fast-JL, which has rigorous theoretical guarantees, this paper provides no theoretical validation for its method. It fails to prove why or how well this heuristic preserves inner products, what its error bounds are, or how it compares to Fast-JL theoretically.
3. Flawed and Invalid Experimental Design
Given that the sole innovation is a compression "plugin," the experiments should have been designed for a component-level comparison, not a method-level comparison. The correct baseline for comparison is Fast-JL, the component DMin aims to replace. A valid experiment would involve: (1) Taking a standard method (e.g., Diffusion-TracIn or D-TRAK) and benchmarking its accuracy, speed, and memory using its original Fast-JL component. (2) Taking the exact same method and replacing its projection module with the proposed DMin component, then running the same benchmark. The paper fails to conduct this critical ablation study. Instead, it presents an invalid comparison between its entire (un-cited) framework and other published methods. Consequently, the experimental results are unconvincing.

**Questions:**

See above

---

### Official Review · Reviewer_Yg3C · 2025-11-05

**Soundness:** 3
**Presentation:** 3
**Contribution:** 3
**Rating:** 6
**Confidence:** 4

**Summary:**

This paper estimates data influence for large diffusion models, a process normally blocked by massive gradient storage requirements. The proposed solution, DMin, uses a first-order approximation and a novel gradient compression technique to shrink storage significantly. These compressed gradients are pre-computed and stored in a k-NN index, allowing for fast retrieval of the most influential training data.

**Strengths:**

The proposed method actually solves a real problem of diffusion model data attribution -- scaling gradient-based methods to large model parameter sizes. The method is simple and easy to use. It is also the first scalable solution to the extend of my knowledge.

**Weaknesses:**

The method has been previously proposed and already adopted in LLM data attribution to demonstrate its effectiveness for large models. The re-application to diffusion models is not exactly novel (though novel in this domain). Some more effort can be put into adapting the technique for the unique diffusion model setting.

**Questions:**

* Table 2 might make a bit more sense if recall@k is reported instead of average@k.
* The method might benefit from adapting it more for the diffusion model setting. Currently it is merely a direct reapplication of OPORP. One direction of exploration might be to investigate the aggregation of influence across time dimensions.

---

### Note · Authors · 2025-11-12

I have read and agree with the venue's withdrawal policy on behalf of myself and my co-authors.